# Postharvest Drying Techniques Regulate Secondary Metabolites and Anti-Neuroinflammatory Activities of *Ganoderma lucidum*

**DOI:** 10.3390/molecules26154484

**Published:** 2021-07-25

**Authors:** Nooruddin-bin Sadiq, Da-Hye Ryu, Jwa-Yeong Cho, A-Hyeon Lee, Dae-Geun Song, Banzragch Dorjsembe, Jin-Chul Kim, Je-Hyeong Jung, Chu-Won Nho, Muhammad Hamayun, Seung-Hoon Yang, Ho-Youn Kim

**Affiliations:** 1Smart Farm Research Center, Korea Institute of Science and Technology (KIST), Gangneung 25451, Korea; nooruddin@kist.re.kr (N.-b.S.); dahye0507@kist.re.kr (D.-H.R.); chocho7023@kist.re.kr (J.-Y.C.); 618002@kist.re.kr (B.D.); jhjung@kist.re.kr (J.-H.J.); cwnho@kist.re.kr (C.-W.N.); 2Department of Plant Science, Gangneung-Wonju National University, Gangneung 25457, Korea; 3Division of Bio-Medical Science and Technology, KIST School, Korea University of Science and Technology (UST), Daejeon 34113, Korea; 4Department of Medical Biotechnology, College of Life Science and Biotechnology, Dongguk University, Seoul 04620, Korea; skandkgus@dgu.ac.kr; 5Natural Products Informatics Center, Korea Institute of Science and Technology (KIST), Gangneung 25451, Korea; dsong82@kist.re.kr (D.-G.S.); jckim@kist.re.kr (J.-C.K.); 6Department of Botany, Abdul Wali Khan University Mardan, Mardan 23200, Pakistan; hamayun@awkum.edu.pk

**Keywords:** *Ganoderma lucidum*, ganoderic acid, neuro-degradation, LPS-induced inflammation, MAPK, BV2 cancer cells

## Abstract

*Ganoderma lucidum* extract is a potent traditional remedy for curing various ailments. Drying is the most important postharvest step during the processing of *Ganoderma lucidum.* The drying process mainly involves heat (36 h at 60 °C) and freeze-drying (36 h at −80 °C). We investigated the effects of different postharvest drying protocols on the metabolites profiling of *Ganoderma lucidum* using GC-MS, followed by an investigation of the anti-neuroinflammatory potential in LPS-treated BV2 microglial cells. A total of 109 primary metabolites were detected from heat and freeze-dried samples. Primary metabolite profiling showed higher levels of amino acids (17.4%) and monosaccharides (8.8%) in the heat-dried extracts, whereas high levels of organic acids (64.1%) were present in the freeze-dried samples. The enzymatic activity, such as ATP-citrate synthase, pyruvate kinase, glyceraldehyde-3-phosphatase dehydrogenase, glutamine synthase, fructose-bisphosphate aldolase, and D-3-phosphoglycerate dehydrogenase, related to the reverse tricarboxylic acid cycle were significantly high in the heat-dried samples. We also observed a decreased phosphorylation level of the MAP kinase (Erk1/2, p38, and JNK) and NF-κB subunit p65 in the heat-dried samples of the BV2 microglia cells. The current study suggests that heat drying improves the production of ganoderic acids by the upregulation of TCA-related pathways, which, in turn, gives a significant reduction in the inflammatory response of LPS-induced BV2 cells. This may be attributed to the inhibition of NF-κB and MAP kinase signaling pathways in cells treated with heat-dried extracts.

## 1. Introduction

Drying is the method of removing the water content by deploying heat. It helps to preserve fresh food products by the suppression of water-linked activities. The process of food drying can be categorized into two major phenomena, i.e., in-vacuum or in-air, based on the medium amount used for dehydration. During drying, the heat can be transferred through different modes, such as radiation, convection, microwave, conduction, and solar sources. In the drying process, the transfer of heat from source to plant material and transfer of water and moisture content from plant material to the surface and surroundings take place simultaneously [1].

In the postharvest process, drying is considered as one of the foremost energy-intensive operation. It has been applied to food products such as fruits, herbal products, and other agricultural products after harvest [2]. Reduced water contents in bio origin products help to improve the shelf life, reduce the growth and development of microbes, lower the enzymatic activities, and inhibit other deteriorative reactions [3]. Some herbal medicines of plant origin need to be dried before extraction of the active constituents [1,4,5].

Any postharvest processing method applied to foods that employ high temperatures may lead to the loss of important heat-sensitive metabolites. The use of a product-adaptive drying method is a very important step in postharvest processing. Antioxidant components such as tocopherols, ascorbic acid, carotenoids, and plant phenolics are highly sensitive to heat, and therefore, freeze-drying is generally recommended [4]. The freeze-drying process involves the gentle removal of water by dehydration through the transformation of liquid contents in food material into ice without increasing the temperature above a normal room temperature. Freeze-drying is considered as the most efficient and effective method in preventing the loss of essential phytochemical compounds and nutrients in dried plant materials as compared to conventional drying methods, like sun, shade, and room temperature drying [5]. However, some studies showed that, as a result of freeze-drying, food materials showed a loss of essential nutritional contents, such as vitamins and minerals [6,7]. Low antioxidant activity is also reported in fruits dried by freeze-drying [8]. With an increase in demand of efficient and cheap drying techniques, heat-drying technologies have gained a lot of R&D efforts to develop cheaper and convenient methods to dry plant products. The materials to be dried can come in various forms and shapes, such as wet solids, pastes, solutions, or suspensions, which need to be dehydrated up to certain point where it can retain its bioactive contents. However, most of the compounds found in plant materials are sensitive to temperature and tend to degenerate, harden, discolorize, or denature when processed through heat-drying techniques [9].

Freeze-drying and conventional heat-drying methods are considered as the most efficient and effective when it comes to use in phytochemical-related industries [10,11]. Other types of dryer-using industries are vacuum dryers, tray dryers, spray dryers, fixed-bed dryers, rotary dryers, fluidized bed dryers, etc. [12].

*Ganoderma lucidum* (GL) is one of the most important species of the family Ganodermataceae and is commonly known as Lin-Zhi in China, Reishi in Japan, and Yeongji in Korea [13]. GL is widely known as the “mushroom of immortality” and is a symbol of traditional Chinese medicine owing to its high therapeutic potential and efficacy [14,15,16]. GL has been used in herbal medicine for many years to treat human diseases, including cancer, viral hepatitis, and bronchitis [17]. Previous studies have reported that the *Ganoderma lucidum* extract (GL extract) could elicit innate immune responses, control cell proliferation, and cause cancer cell migration [18,19,20]. Metabolite profiling showed that ganoderic acids and polysaccharides are the main components of GL responsible for its efficacy [14]. The most conventional methods used for drying GL are heat-drying and freeze-drying in the industry [21].

Inflammation is a hallmark of many neurodegenerative diseases, and microglia regulate the immune responses in the central nervous system (CNS) [22]. The inflammatory process is initiated by the innate immune system in order to protect the brain and accompanying neurons [23]. The presence of chronic inflammation may lead to the overproduction of proinflammatory cytokines that play a significant role in the development of neurodegenerative diseases such as Alzheimer’s disease (AD) and Parkinson’s disease (PD) [24]. The inflammatory response is triggered by pathogen-associated molecular patterns (PAMPs) and damage-associated molecular patterns (DAMPs), such as lipopolysaccharides (LPS) and viruses [25]. In the overall step of inflammation, many cytokines are produced and secreted by various types of immune cells via nuclear factor kappa B (NF-κB) and MAP kinase (MAPK) signaling pathways [26].

MAPK engages in biological responses, including cell proliferation, differentiation, and death. In MAPK signaling, three successively activated protein kinases called p38, c-Jun N-terminal kinase (JNK), and extracellular signal-regulated kinase (ERK) are involved. The JNK and p38 modules activated by inflammatory cytokines or environmental stresses play an essential role in inflammation [27]. ERK controls the production of inflammatory cytokines such as tumor necrosis factor-alpha (TNF-α) [28]. Recently, several studies have reported the critical roles of NF-κB and MAPK signaling in the progression of neurodegenerative diseases such as PD and AD [29,30]. Therefore, the discovery of potential candidates regulating the secretion of proinflammatory cytokines by microglia is essential for the treatment of neurodegenerative diseases [31].

Due to the pharmaceutical significance of GL, its fruiting bodies are cultivated on wood logs at the industrial level. During the postharvesting period, drying is considered as the most energy- and time-consuming process. The fruiting bodies are dried to enhance their shelf lives, make transportation easy, and prevent the degradation of the important bioactive compounds. In our previous study, extraction optimization and ganoderic acid content determination in heat- and freeze-dried GL fruiting bodies were performed, which showed differences in the secondary metabolite contents of the fruiting bodies dried by the two methods [32]. Several studies that have employed freeze-drying [33] used methanolic extract [34] and determined the polysaccharide level [35] of GL demonstrated that the GL extract reduced the production of proinflammatory cytokines and inflammatory signaling proteins in LPS-treated immune cells such as macrophages. However, little is known about the metabolic constituents and the effect of the optimized ethanolic extract of heat-dried GL on neuroinflammation.

This study was performed as a continuation of our previous study on the extraction optimization and detection of ganoderic acid in both heat- and freeze-dried GL samples. In the present study, we determined the effects of two different (heat-drying and freeze-drying) postharvest drying methods used commercially in metabolite profiling and their therapeutic effects. GC-MS-based However, little is known about the metabolic constituents and the effect of the optimized ethanolic extract of heat-dried GL on neuroinflammation.

## 2. Results

### 2.1. Primary Metabolites as Affected by Different Drying Methods

Ganoderma lucidum shows changes in metabolite profiling with the developmental stages, the mature stage being the most stable stage in terms of primary metabolite profiling [36]. A GC-TOF-MS-based analysis was performed using the liquid-phase extraction method to detect the primary metabolites in heat- and freeze-dried samples. A total of 109 compounds were detected in a crude GL extract, as summarized in Appendix A, of which 31 showed significant differences between heat- and freeze-dried samples based on the analysis of variance (*p* < 0.05) (Table 1). Among the heat-dried and freeze-dried samples, the heat-dried samples contained organic acids (50.4%), alcohols and derivatives (18.3%), amino acids (17.4%), and carbohydrates (8.8%), whereas, in freeze-dried samples, organic acids (64.1%), alcohol and its derivatives (9.14%), amino acids (7.5%), and carbohydrates (5.4%). Freeze-dried samples showed similar results, as already reported by Guo et al. [37]. During the freeze-drying process, the metabolisms of organic acids decrease due to a decrease in the citrate and malate contents [38]. A reduced dataset of 53 metabolites common in both the treatments was used to perform the PC analysis (Figure 1). A total of 77.5% variance was explained in both the components (PC1: 63.4% and PC2: 14.1%) by separating the metabolite profiles of the samples subject to both the treatments. PC1 successfully captured the variance in metabolites during the drying process. The PLS-DA biplot accurately distinguished the heat-dried samples from the freeze-dried ones into two groups. PC1 separated the heat- and freeze-dried samples based on the main changes in amino acid, organic acid, and the monosaccharide contents. The heat-dried extract was differentiated due to a high level of amino acids (L-Alanine, L-Serine, L-Asparagine, L-Proline, and L-Threonine); alcohols and derivatives (D Mannitol, glycerol, ethylene glycol, and silanol) and monosaccharides (D-(−)- fructose, D-(+)- mannose and D-(−)- ribofuranose). In the freeze-dried samples, the organic acids were found to be significantly higher.

### 2.2. Proteomic Analysis

The heat-treated samples showed higher concentrations of ATP-citrate synthase (ACS), pyruvate kinase, glyceraldehyde-3-phosphatase dehydrogenase, glutamine synthase, fructose-bisphosphate aldolase, and D-3-phosphoglycerate dehydrogenase than the freeze-dried samples (Figure 2A). The total peak area of ACS was 17.86 ± 3.4 in the heat-dried samples; in the freeze-dried samples, ACS was not detected. Pyruvate kinase was the second-most abundant detected enzyme in the heat-dried samples, with a peak area of 13.31 ± 3.21; in the freeze-dried samples, the total peak area for the pyruvate kinase was 6.09 ± 2.41. The catalase (CAT), ATP synthase, and phosphoglycerate kinase (PK) 1 levels were found to be higher in the freeze-dried samples (Figure 2A). Catalase had the highest peak detected in the freeze-dried samples, with total area of 56.11 ± 3.64 when compared to the heat-dried samples; the total peak area was 38.21 ± 4.7. The peak area of the ATP synthase was 42.02 ± 6.29 in the freeze-dried samples, which was higher when compared to the heat-dried samples: 22.55 ± 0.24.

About 78.4% of the variance was explained by the PC1 and PC2 components based on the PCA analysis performed on a reduced dataset of 20 significant enzymes detected in the protein analysis. PC1 showed the highest values for ATPase, PK1, CAT, chorismite synthase (CS), and ACS and showed the lowest value for cystathionine beta-synthase (CBS) (Figure 2B). Succinic acid dehydrogenase (SDADH), PK, and isopentenyl diphosphate (IPP) isomerase showed the highest values, while glyceraldehyde-3-phosphate dehydrogenase (GAPDH), pyruvate carboxylase (PC), and the proteasome endopeptidase complex (PEC) showed the lowest values for PC2. These enzymes had the highest impacts on the clear separation of heat-dried samples from the freeze-dried samples. The PCA analysis clearly showed that the heat-dried samples exhibited differences in the protein contents compared with the freeze-dried samples.

### 2.3. Cytotoxicity Assay

To investigate the effect of each GL extract on the physiological activities in neuronal cells, we performed cell viability assays using BV2 cells. The cells were treated with a GL extract in a dose-dependent manner (1–100 μg/mL). We observed that low concentrations (1 μg/mL) of both the GL extract samples did not significantly affect the cytotoxicity. In some GL extracts with high concentrations (>10 μg/mL), a mild cell toxicity was recorded (Figure 3). This result indicated that the GL extracts did not have any cytotoxic effects on the neural cell line with low concentrations.

### 2.4. Heat-Dried GL Extract Inhibits the Activation of NF-κB Signaling in LPS-Treated BV2 Cells

The MAPK signaling pathway is a key mediator in activating the production of proinflammatory cytokines secreted by the microglia [39,40]. To determine whether heat- or freeze-dried GL extracts can affect the activation of inflammatory mediators through MAPKs, their effects on phosphorylated MAPKs in LPS-treated microglia were determined. A Western blot analysis showed that the heat-dried GL extract dramatically inhibited the LPS-induced phosphorylation of the representative MAPKs, including p38, JNK, and ERK, in a dose-dependent manner (Figure 4A–D) when compared with the freeze-dried GL extract, which had no significant effect on the phosphorylation of the MAPKs (Figure 5A–D).

To determine whether the heat-dried GL extract or freeze-dried GL extract had a suppressive effect on the activation of the NF-κB signaling pathway, the level of p65 phosphorylation (a major molecular component of NF-κB signaling) was determined in the LPS-treated microglial cell line (BV2 cell line). We found that the heat-dried GL extract significantly inhibited the LPS-induced phosphorylation of p65 in this cell line in a dose-dependent manner (Figure 4E,F). The freeze-dried GL extract showed no significant inhibitory effect on the p65 phosphorylation induced by LPS (Figure 5E,F). Therefore, the heat-dried GL extract inhibits the activation of NF-κB signaling in LPS-treated BV2 cells. The inhibition of nuclear translocation of NF-κB p65 by the heat-dried GL extract may be the reason for the regulation of the neuroprotective events.

## 3. Discussion

In this study, we determined the effects of two commonly used drying methods on the metabolites profiling and anti-neurodegenerative potential of the GL extract in LPS-treated microglia via NF-κB signaling and MAPK signaling. In NF-κB signaling, the effect of the heat-dried GL extract was concentration-dependent. As the GL extract concentration increased, the expression of phospho-p65 was inhibited. Moreover, the GL extract also had a dramatic effect on the activation of MAPKs, including p38, JNK, and ERK.

GL is used as an herbal remedy for many ailments in Eastern Asia [38,39,40]. The GL extract has therapeutic benefits and is used for the treatment of diabetes, breast cancer, and inflammatory diseases such as colitis [40] Previous studies have reported on its neuroprotective effects that can be useful for the prevention of AD and PD [41].

We studied the variations in GL primary metabolites by applying two different drying temperatures. Heat-drying may change the activity of certain enzymes after harvesting [41]. On the other hand, freeze-drying at a low temperature assisted by a vacuum treatment helps keep biological molecules intact during the entire drying process [42,43,44] and also preserves the primary and secondary metabolite contents [45]. In GL, different drying methods can yield different biological products under similar conditions. A protein analysis by SDS-PAGE provided significant insights into the observed differences and enabled us to study the inhibitory effects of heat- and freeze-dried GL extracts on LPS-treated microglia via NF-κB and MAPK signaling. Our primary data analysis showed that, during heat-drying, more carbohydrates, amino acids, and ganoderic acids were produced compared to freeze-drying. During heat-drying, the biosynthesis of amino acids and monosaccharides was high, followed by a high ganoderic acid content. Higher amino acid contents can be attributed to the degradation of proteins at high temperatures, as evident from our data (Table 1 and Figure 1).

In fungi, tricarboxylic acid (TCA) and glyoxylate cycles are important energy-generation pathways comprising two or three carbon compounds. Under limited nutrients, fungi start depending on compounds with higher molecular weights, such as fatty acids, and secondary metabolites by activating TCA and glyoxylate cycles [46]. During the heating process, GL activates alternative carbon pathways using different carbon sources instead of glucose to fulfill the energy demand [47]. Pyruvate/glyceraldehyde-3-phosphate and mevalonate pathways lead to the biosynthesis of ganoderic acids. In eukaryotes, acetyl-CoA serves as a precursor for the production of dimethylallyl diphosphate and IPP through the mevalonate pathway [48]. In our study, an increase in the primary metabolite content (especially monosaccharides) suggests the activation of the glyoxylate cycle, followed by gluconeogenesis. The activation of the glyoxylate cycle suggests that the cell uses lipids as a primary source to generate ATP by the β-oxidation of fatty acids to acetyl-CoA. The acetyl-CoA is then channeled through the glyoxylate cycle and used by malate synthase to produce malate [49].

Our proteomics study showed that the levels of glyoxylate cycle-related enzymes in GL were increased in response to heat (Figure 2), while the levels of enzymes such as SDADH malate dehydrogenase (MDH) were decreased. However, the levels of enzymes such as GAPDH, PK1, ACS, and PK that are directly involved in the acetyl-CoA synthesis pathway were increased in GL in response to heat. GAPDH and PGK are pivotal for the conversion of fructose/glucose to pyruvate, which can be converted to glucose through gluconeogenesis and returned to phosphoenolpyruvate, which is generated from pyruvate by a catalytic reaction of the pyruvate kinase [50]. PK catalyzes the conversion of pyruvate to acetyl-CoA [51].

Ganoderic acids exert anti-inflammatory activity [52,53], antidiabetic activity [54], anticancer activity [55], and antitumor activity [56,57] and is useful in prostate hypertrophy treatment [58]. In our previous studies, we reported that ganoderic acids B, H, K, and G and lucidenic acid N were significantly high in the heat-dried samples, while 20-hydroxyganoderic acid AM1; ganoderenic acids D and K; ganoderic acids A, D, and F; and lucidenic acid D were significantly high in the freeze-dried samples [32]. In this study, we observed an increase in the monosaccharide and amino acid contents and levels of the enzymes responsible for the conversion of malate to pyruvate and acetyl-CoA to citrate and oxaloacetate, along with an increase in the ganoderic acid contents in some heat-dried samples. In previous studies, the ability of GL to produce ganoderic acid was affected by different culture conditions, and the effect of temperature on the ganoderic acid content was the highest among the various factors [59,60]. This suggests that, during heat-drying, the lack of oxygen activates the selective enzymes of the glyoxylate cycle, followed by gluconeogenesis, which results in the production of more carbohydrates and precursor molecules for the biosynthesis of ganoderic acids. Acetyl-CoA also serves as a precursor molecule for the biosynthetic pathway of ganoderic acids. Moreover, the enzymes related to the triterpenoid synthesis pathway have also been reported to be determined by amino acids [61]. Therefore, an increase in acetyl-CoA activates the ganoderic acid biosynthesis pathway via the activation of some other specific pathways based on increased amino acid production under the drying method.

Neuroinflammation is the first response to the onset and progression of neurodegenerative diseases like PD and AD. The inflammation of neurons is actively involved in the progression of disease. Neuroinflammation is a protective response in the case of injuries to neurons, and it helps to mediate and restore damaged glial and neuronal cells. However, in the chronic stage, it causes degradation of the neuronal cells. The NF-ĸB response is different at different stages of disease progression in AD; it is involved in the development of plaque and during the signaling of cytokine and inflammation during disease progression. Slowing down the activity of NF-ĸB results in decreased progression of the disease [34].

We observed that, in NF-κB signaling, the effect of the GL extract was dependent on the concentration of the GL extract. As the concentration increased, the expression of phospho-p65 was inhibited. Moreover, the GL extract also had a inhibitory effect on the activation of MAPK, including p38, JNK, and ERK [34]. We observed that the heat-dried GL extract had a significantly higher inhibitory effect on NF-κB and MAPK signaling compared to the freeze-dried GL extract, indicating that the inhibitory effect of the GL extract is dependent on the postharvest treatment.

## 4. Materials and Methods

### 4.1. General Procedure

GL samples were procured from Korea Ginseng Corporation (Daejeon, Korea). Heat- and freeze-dried samples were dehydrated for 36 h at 60 °C and −80 °C, respectively. For metabolite profiling, the freeze-dried samples (10 mg) were dissolved in the extraction solvent mixture of water, methanol, and chloroform (water:methanol:chloroform = 1:2.5:1, *v/v/v*), including ribitol and 5α-cholestane as the internal standard. Extraction was performed at 37 °C, and the samples were mixed at 1200 rpm for 30 min. The extracted solution was centrifuged for 3 min at 13,000 rpm. The polar phase was collected into a new tube, and distilled water was added. Samples were centrifuged at 13,000 rpm for 3 min, and the methanol/water phases were dried using a freeze dryer. The derivative process was performed using methoxyamine hydrochloride (methoxime) and trimethylsilyl [62].

### 4.2. Gas Chromatography Coupled with Time-of-Flight Mass Spectrometric (GC-TOF-MS) Analysis

An analysis of the primary metabolites was performed using GC-TOF-MS (LECO Pegasus GC HRT, Leco Corp., St. Joseph, MI, USA). The sample was injected in a split mode (1:25) with an injection volume of 1 μL at 230 °C. Carrier gas (helium) flowed at a rate of 1 mL min^−1^. The column used was Rtx-5MS (30 m × 0.25 mm; film thickness, 0.25 μ). The starting temperature of the oven was 80 °C for 2 min and was then increased by 15 °C per min to attain 320 °C for 10 min. The transfer line temperature was programmed to be 250 °C, and the ion source temperature was 200 °C. TOF-MS acquisition was delayed for 180 s, with an acquisition rate of 20 spectra per second. The detector voltage and electron energy were set at 1650 V and 70 eV, respectively [62].

### 4.3. Protein Extraction

Approximately 40 mg of powdered GL samples were transferred to a prefilled bead tube (dry garnet, 0.7 mm, #1103439, Qiagen, Hilden, Germany). A 500-µL Lysis buffer (50-mM Tris-Cl, pH 8.0, and 5-mM EDTA), 0.4% sodium lauryl sulfate (SDS), 50-mM NaCl, and 1-mM PMSF) was added to the tube and homogenized at 30 pulses/s for 8 min, with a TissueLyser II (Qiagen, Hilden, Germany). After heating the samples at 60 °C for 15 min, the samples were sonicated (Vibra-cell™, Sonics, Newtown, CT, USA) at 30% amplitude for 2-s and 1-s on/off cycles, respectively, for 1 min. The sonicated samples were centrifuged at 13,000 rpm at 4 °C for 15 min, and the supernatants were transferred to a new tube. A bicinchoninic acid (BCA) gold protein assay kit (Thermo Fisher Scientific, San Jose, CA, USA) was used for the quantification of the crude protein. The crude protein samples (100 µg) were precipitated using 5 volumes of cold acetone and stored at −20 °C overnight. After centrifugation at 13,000 rpm, 4 °C for 15 min, the protein precipitates were collected. The pellets were resuspended in a LDS sample buffer (Thermo Fisher Scientific, San Jose, CA, USA) containing 50-mM DTT to a final concentration of 2 mg mL^−1^ and heated at 95 °C for 10 min [63].

### 4.4. Peptide Analysis

The crude protein samples (40 µg) were separated on a NuPage Bis-Tris SDS-PAGE gel (4–12%) (Life Technologies, Grand Island, NY, USA). Coomassie brilliant blue g-250 was used as a staining agent. The gels were subjected to a conventional in-gel digestion with minor modifications (Kang et al. 2014). The Q Exactive™ Hybrid Quadrupole-Orbitrap^™^ Mass Spectrometer (Thermo Scientific, San Jose, CA, USA) equipped with an Easy-nLC 1000 system (Eksigent, Dublin, CA, USA) was used for the peptide analysis. A 5-μL peptide sample was injected and loaded onto an Acclaim PepMap^™^ 100 trap column (2 cm × 75 μ; Thermo Scientific, San Jose, CA, USA). An EASY-Spray column (15 cm × 50 μ ID; particle size, 2 μ; pore size, 100 Å; PepMap™ RSLC C18, Thermo Scientific, San Jose, CA, USA) was used for sample separation. Solvent A (0.1% formic acid in water) and B (0.1% formic acid in ACN) at a flow rate of 300 nL min^−1^ were used. The gradient started with 5% solvent B for 5 min and increased to 50% by 40 min and then increased to 100% by 41 min and was maintained for another 5 min. Prior to sample loading, the trap and analytical columns were equilibrated with a 10× column volume of solvent A. The eluted peptides were ionized with a voltage of 2 kV and subjected to mass spectrometry. The peptide ions were first analyzed with a full MS scan in a range of 50–2000 *m/z* and a resolution of 70,000.

Tandem mass spectra were analyzed using the SEQUEST module of Proteome Discoverer (Thermo Fisher Scientific, San Jose, CA, USA; version 1.4.1.14). The keyword “organism: Ganoderma” (downloaded on 26 September 2019; a total of 16,289 entries) was used to download the protein database from UniProt. Scaffold (version Scaffold_4.9.0, Proteome Software Inc., Portland, OR, USA) was used to validate the MS/MS-based peptide and protein identifications. Normalized spectrum counting was performed for the semi-quantitative analysis of the proteins [64].

### 4.5. Cell Viability Assays

The cytotoxicity of each extract in the BV2 cells was evaluated by the MTT assay (Promega) by following the manufacturer’s instructions. BV2 cells (1 × 10^4^ cells/well) were seeded into a 96-well plate. BV2 cells were treated with each extract for 12 h. After adding a solubilization solution and incubating the mixture at room temperature for 16 h, the insoluble formazan was measured at 570 nm [65].

### 4.6. LPS Treatment to BV2 Cells for Inflammation Induction

BV2 cells were seeded into a 6-well plate. The number of cells seeded per well of the plate was 2 × 10^6^ cells. Cell starvation was induced by decreasing the concentration of FBS to 0.5%. BV2 cells were then pretreated with different concentrations of the GL extract (0.4, 2, and 10 µg/mL) for 15 min at 37 °C. BV2 cells in each well were cotreated with LPS (1 µg/mL), followed by the incubation at 37 °C for 12 h. One milliliter of supernatant was taken. The cells in each well were washed with cold 1X phosphate buffer solution, followed by the treatment with tissue lysis buffer containing 1% SDS (200 µL). Then, the cell lysate was scraped and collected in an e-tube [65].

### 4.7. Western-Blot Analysis

A BCA assay was used to measure the protein concentration of cell lysates. The solution was separated by SDS-polyacrylamide gel electrophoresis (PAGE) and then transferred to a protein membrane and blocked with 5% skimmed milk in 1× phosphate-buffered saline with 2% Tween-20. The cells were then treated overnight at 4 °C with the primary antibodies (1:1000) and then with the secondary antibodies (1:10,000). The maximum sensitivity substrate and signal intensities were measured [65].

### 4.8. Statistical Analysis

The software package SIMCA-P 15.0.2 (Umetrics, Umeå, Sweden) was used for the multivariate data analysis. A principal component analysis (PCA) was performed with auto-scaled data to understand an overview of the group clustering and to identify the possible outliers. Data were presented as the mean of the standard deviation. All the experiments were performed in triplicate.

## 5. Conclusions

Among the two commonly used drying techniques we investigated, the heat-drying method was found to be the most effective method in the postharvest processing of Ganoderma lucidum. As compared to the freeze-drying method, the therapeutic potential was affected to a greater degree. This suggests that the postharvest drying process has a greater impact on the pharmacological properties of Ganoderma lucidum.

The results showed that heat-drying helps to improve the metabolites content by the upregulation of the enzymes related to the TCA cycle. The heat-dried extract also performed very well against LPS-induced neuroinflammation in the BV2 cells. This suggests that the heat-dried GL extract can be used as an important drug component in slowing the progression of neurodegenerative diseases.

This provides strong evidence that the postharvest drying methods can be a decisive factor in the selection of raw materials for the pharmaceutical industry. The heat-dried GL extract may play a critical role in the development of potential drugs for the neurodegenerative diseases resulting from LPS-induced neuroinflammation.

## Figures and Tables

**Figure 1 molecules-26-04484-f001:**
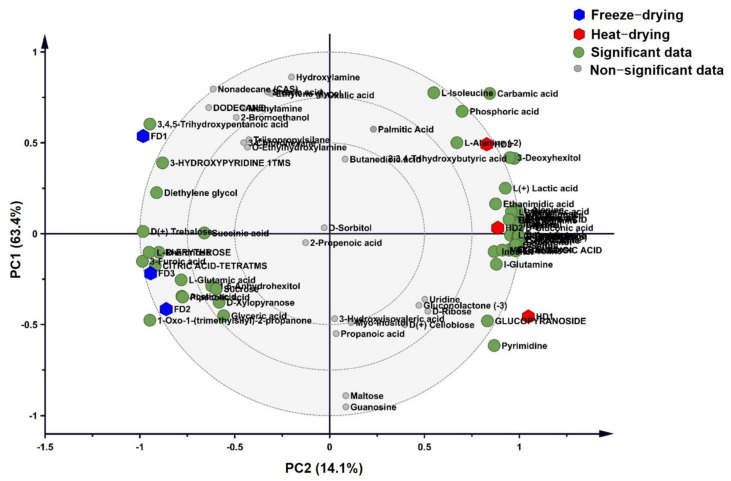
A principal component analysis of the primary data detected by GC-TOF-MS was performed in the SIMCA program. For the metabolite analysis, three biological replicates per treatment sample were analyzed, and the results were presented in principal component 1 (PC1, 63.4%) and PC2 (14.1%). The differences in the primary metabolite data between freeze-drying (blue hexagon) and heat-drying (red hexagon) in the Student’s t-test are marked as green circles (*p* < 0.05), and the nonsignificant data are indicated by gray circles (*p* > 0.05).

**Figure 2 molecules-26-04484-f002:**
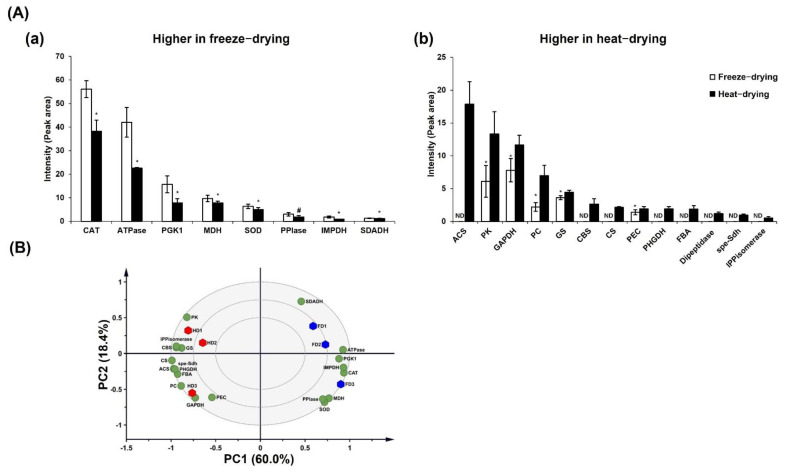
Protein analysis based on SDS-PAGE-LC-MS. (**A**) Proteins were significantly (*p* < 0.05) changed between freeze-drying (**Aa**) and heat-drying (**Ab**) in the Student’s *t*-test and classified based on the drying methods in the PCA biplot (*: *p* < 0.05 and #: *p* < 0.005). (**B**) Blue hexagons represent the freeze-dried samples, red hexagons represent the head-dried samples, and the significant protein molecules are expressed by green circles. The results are expressed as the mean ± standard deviation, and the *t*-test was performed to confirm the significant differences. PC: principal component; ND: not detected; CAT: catalase; ATPase: ATP synthase; PGK1: phosphoglycerate kinase1; MDH: malate dehydrogenase; SOD: superoxide dismutase; PPlase: peptidyl-prolyl cis-trans isomerase; IMPDH: Inosine-5′-monophosphate dehydrogenase; SDADH: succinate semialdehyde dehydrogenase; ACS: ATP-citrate synthase; PK: pyruvate kinase; GAPDH: glyceraldehyde-3-phosphate dehydrogenase; PC: pyruvate carboxylase; GS: glutamine synthetase; CBS: cystathionine beta-synthase; CS: chorismite synthase; PEC: proteasome endopeptidase complex; PHGDH: D-3-phosphoglycerate dehydrogenase; FBA: fructose-bisphosphate aldolase; spe-Sdh: Spermidine synthase-saccharopine dehydrogenase; IPPisomerase: Isopentenyl diphosphate isomerase.

**Figure 3 molecules-26-04484-f003:**
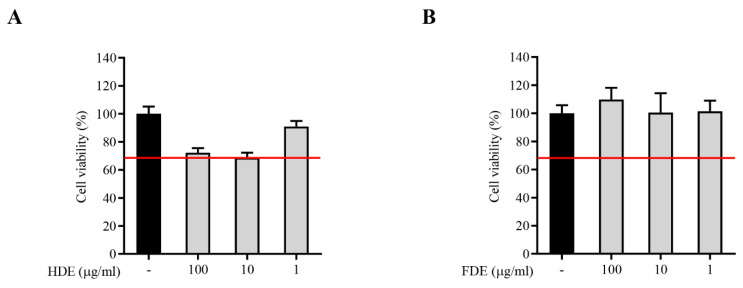
The effects of the heat-dried extract (HDE) (**A**) and freeze-dried extract (FDE) (**B**) on the cell viability in microglial BV2 cells were assessed by the MTT assay. The BV2 cells were treated with various concentration (1, 10, and 100 µg/mL) of GL extracts, and the cell viability was measured. The red line indicates a 70% acceptable range of cell viability.

**Figure 4 molecules-26-04484-f004:**
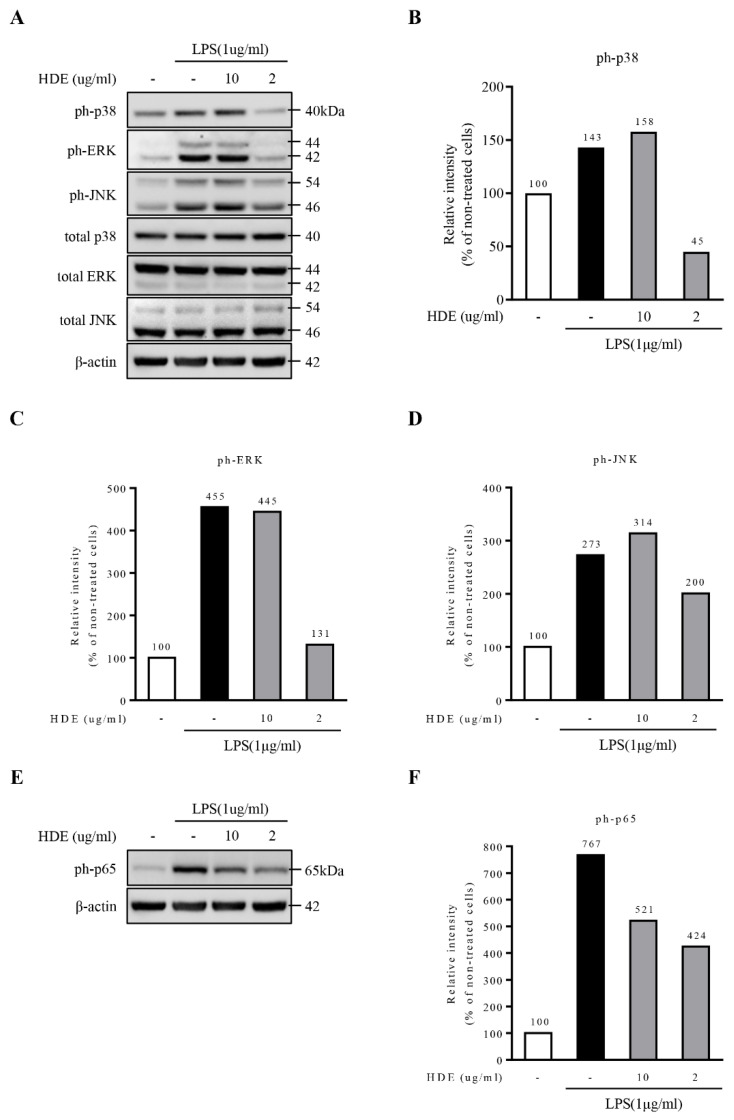
Inhibitory effect of HDE on the MAPKs in LPS-stimulated BV2 cells. (**A**–**D**) Western blot analysis on the p38, ERK1/2, and JNK phosphorylation levels in the BV2 cells. The cells were pretreated with HDE 2–10 µg/mL for 2 h prior to LPS (1 µg/mL) exposure for 24 h. (**E**,**F**) Western blot analysis on ph-p65 and protein expression in the BV2 cells when pretreated with HDE prior to LPS exposure. β-actin was used as the housekeeping protein.

**Figure 5 molecules-26-04484-f005:**
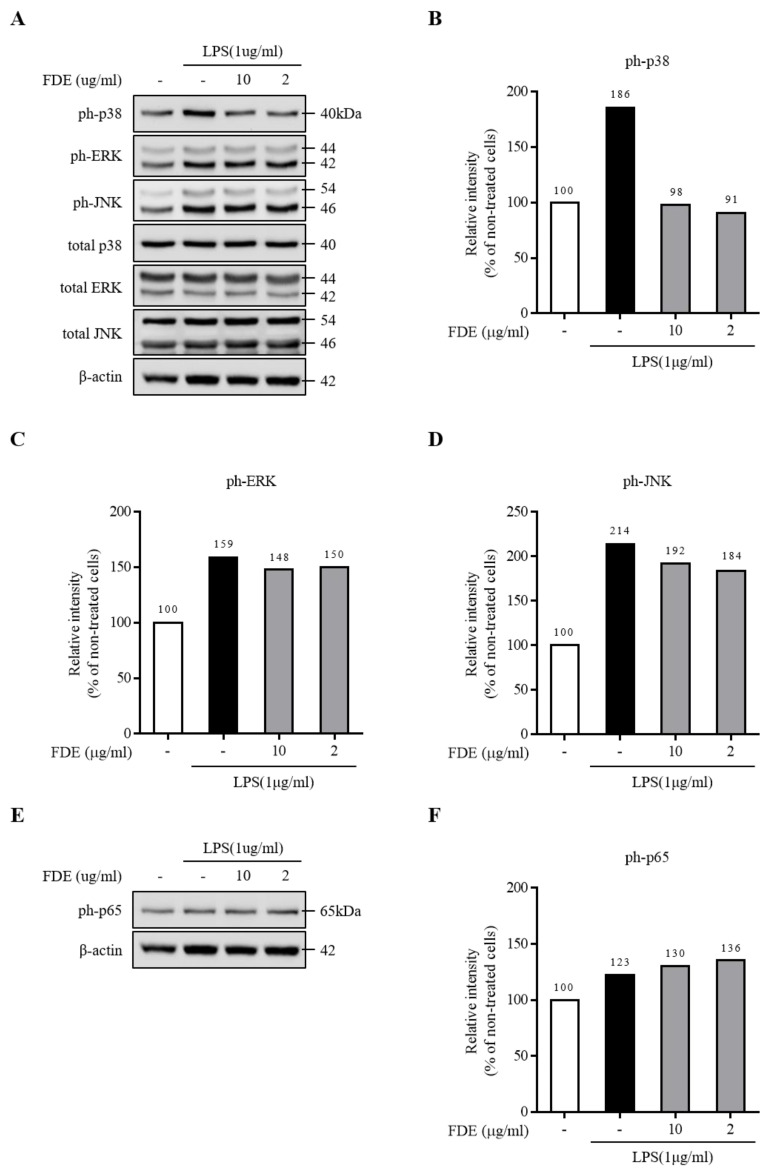
(**A**–**D**) Effect of the freeze-dried extract (FDE) on the phosphorylation levels of ph-p65, ph-p38, and ph-JNK in LPS-treated BV2 cells pretreated with FDE 2–10 µg/mL for 2 h, followed by 24 h of exposure to LPS (1 µg/mL). The cells were lysed with a radioimmunoprecipitation assay buffer, and the phosphorylation levels were measured by performing an immunoblot analysis. (**E**,**F**) Western blot analysis on ph-p65 and the protein expression in BV2 cells when pretreated with FDE prior to LPS exposure. β-actin was used as the housekeeping protein.

**Table 1 molecules-26-04484-t001:** Significantly different primary metabolites between heat-dried (60 °C) and freeze-dried (−80 °C) samples by GC-TOF-MS (*p* < 0.05). Three biological replications per each treatment sample were analyzed. ND means “not detected”.

No	RT (s)	Identified Compound	Formula	S/N	Temp (°C)	Peak Area	*p*-Value
Organic acid							
1	336.475	L-(+)-Lactic acid	C_3_H_6_O_3_	259.64 ± 46.50	−80	623,113 ± 220,546	0.005
					60	1,353,365 ± 61,640	
2	388.033	3-Furoic acid	C_5_H_4_O_3_	120.63 ± 49.69	−80	268,829 ± 14,818	0.000
					60	99,835 ± 4186	
3	495.075	Succinic acid	C_4_H_6_O_4_	843.21 ± 31.93	−80	6,780,625 ± 1,330,300	0.000
					60	ND	
4	590.617	Malic acid	C_4_H_6_O_5_	5741.99 ± 23.93	−80	3,368,389 ± 350,517	0.001
					60	4,294,037 ± 67,108	
5	731.117	Propanoic acid	C_3_H_6_O_4_	52.45 ± 2.66	−80	ND	0.000
					60	19,674± 1922	
6	745.183	Citric acid	C_6_H_8_O_7_	10,707.06 ± 39.46	−80	7,468,684 ± 1,296,440	0.009
					60	3,794,228 ±237,974	
7	824.033	D-Gluconic acid	C_6_H_12_O_7_	1322.40 ± 100.12	−80	54,672 ± 16,882	0.000
					60	1,060,360 ± 119,163	
Amino acid/Amine							
8	355.775	L-Valine	C_5_H_11_NO_2_	300.84 ± 94.39	−80	292,186 ± 64,672	0.002
					60	2,214,288 ± 462,985	
9	361.750	N-Butylamine	C_4_H_11_N	500.24 ± 85.16	−80	36,110 ± 538	0.002
					60	798,463 ± 103,414	
10	365.325	L-Alanine	C_3_H_7_NO_2_	2235.52 ± 70.06	−80	2,218,555 ± 620,914	0.000
					60	9,117,948 ± 656,401	
11	439.383	D-Valine	C_5_H_11_NO_2_	5420.60 ± 103.17	−80	475,411 ± 136,233	0.000
					60	12,989,286 ± 224,714	
12	470.525	Ethanolamine	C_2_H_7_NO	568.73 ± 56.24	−80	281,663 ± 94,049	0.000
					60	822,952 ± 83,137	
13	472.700	L-Norleucine	C_6_H_13_NO_2_	2223.83 ± 104.51	−80	167,868 ± 61,864	0.002
					60	6,097,243 ± 864,121	
14	486.208	L-Leucine	C_6_H_13_NO_2_	2240.74 ± 102.79	−80	225,995 ± 70,362	0.001
					60	6,090,595 ± 690,905	
15	490.150	L-Proline	C_5_H_9_NO_2_	1541.72 ± 107.91	−80	66,414 ± 33,100	0.000
					60	4,183,781 ± 920,171	
16	523.325	L-Serine	C_3_H_7_NO_3_	1096.97 ± 90.24	−80	346,714 ± 128,627	0.001
					60	3,385,954 ± 645,120	
17	538.683	L-Threonine	C_4_H_9_NO_3_	821.96 ± 9134	−80	163,522 ± 45,729	0.001
					60	1,784,039 ± 211,063	
18	607.075	L-Aspartic acid	C_4_H_7_NO_4_	2225.12 ± 62.37	−80	622,475 ± 115,745	0.002
					60	2,060,365 ± 6360	
19	653.967	L-Glutamic acid	C_5_H_9_NO_4_	1138.54 ± 49.91	−80	356,869 ± 131,194	0.005
					60	855,723 ± 81,835	
20	662.317	Phenylalanine	C_9_H_11_NO_2_	467.51 ± 86.40	−80	144,403 ± 35,274	0.000
					60	1,228,814 ± 130,812	
21	679.717	L-Asparagine	C_4_H_8_N_2_O_3_	173.74 ± 81.35	−80	112,420 ± 56,004	0.000
					60	727,603 ± 65,606	
22	785.425	L-Lysine	C_6_H_14_N_2_O_2_	726.90 ± 84.47	−80	18,913 ± 41,286	0.000
					60	1,631,107 ± 166,827	
Monosaccharides							
23	709.058	L-Rhamnose	C_6_H_12_O_5_	452.46 ± 101.89	−80	164,194 ± 29,950	0.000
					60	3,186,455 ± 211,877	
24	751.675	D-(-)-Erythorose	C_4_H_8_O_4_	261.62 ± 69.45	−80	7,495,006 ± 2,173,719	0.002
					60	2,337,360 ± 444,077	
25	770.233	D-(-)-Fructose	C_6_H_12_O_6_	196.91 ± 28.92	−80	137,329 ± 25,989	0.003
					60	1,301,930 ± 50,758	
26	781.567	D-(+)-Mannose	C_6_H_12_O_6_	12,225.38 ± 84.45	−80	1,339,283 ± 309,865	0.000
					60	9,066,007 ± 634,819	
27	794.742	D-Mannitol	C_6_H_14_O_6_	18,746.15 ± 103.89	−80	502,455 ± 193,329	0.000
					60	16,399,566 ± 2,209,653	
28	857.492	Inositol	C_6_H_12_O_6_	5254.67 ± 30.80	−80	2,211,218 ± 527,434	0.026
					60	3,361,783 ± 225,679	
29	887.150	D-Sorbitol	C_6_H_14_O_6_	71.30 ± 7.61	−80	ND	0.000
					60	43,967 ± 4705	
30	938.000	D-(-)-Ribofuranose	C_5_H_10_O_5_	54.96 ± 9.83	−80	ND	0.000
					60	134,705 ± 13,507	
31	1070.190	D-(+)-Trehalose	C_12_H_22_O_11_	42,467.71 ± 32.92	−80	19,711,971 ± 726,018	0.000
					60	11,425,742 ± 1,430,771	
Alcohols and its derivatives							
32	748.625	3-Deoxyhexitol	C_6_H_14_O_5_	48.303 ± 40.76	−80	16,294 ± 6668	0.024
					60	44,286 ± 894	
33	794.742	D Mannitol	C_6_H_14_O_6_	18,746.15 ± 103.88	−80	502,454 ± 17,291	0.007
					60	16,399,565 ± 19,763	
34	454.675	Diethylene glycol	C_4_H_10_O_3_	48.874 ± 61.899	−80	408,235 ± 5816	0.05
					60	116,301 ± 6540	
35	887.150	D-Sorbitol	C_6_H_14_O_6_	71.296 ± 7.608	−80	ND	
					60	43,967 ± 420	
36	224.233	Ethylene glycol	C_2_H_6_O_2_	2745.569 ± 8.64	−80	19,748,319 ± 53,976	0.02
					60	17,779,139 ± 9579	
37	472.950	Glycerol	C_3_H_8_O_3_	1073.492 ± 80.425	−80	426,256 ± 10,359	0.001
					60	2,784,540 ± 4976	
38	475.608	Silanol	S_i_H_4_O	28,812.306 ± 56.396	−80	8,028,737 ± 31,479	0.005
					60	27,759,383 ± 2333	
39	597.350	L(-) Arabitol	C_5_H_12_O_5_	16.866 ± 26.507	−80	29,802 ± 1857	0.03
					60	48,490 ± 446	

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
