# Peer review of "Postharvest Drying Techniques Regulate Secondary Metabolites and Anti-Neuroinflammatory Activities of Ganoderma lucidum"

_molecules, 2021, doi:10.3390/molecules26154484_

Round 1

Reviewer 1 Report

The manuscript by Sadiq et al entitled “Postharvest drying techniques regulate secondary metabolites and anti-neuroinflammatory activities of Ganoderma lucidum” describes the effect of the drying method involved on the secondary metabolite contents and the anti-neuroinflammatory activities of the Ganoderma lucidum extract. From my point of view, the manuscript covered most of the points I searched for. However, minor points are to be considered:

  • Add a conclusion section with a recommendation for the use of Ganoderma lucidum.
  • Unify the word post-harvest.
  • Avoid personalization.
  • Give some information about the organism cultivation, identifier, season of collection, …
  • Figure 1 is not clear to the reader.

Author Response

  1. Add a conclusion section with a recommendation for the use of Ganoderma lucidum.

Reply: conclusion section added after discussion.

  1. Unify the word post-harvest.

Reply: Done as suggested

  1. Give some information about the organism cultivation, identifier, season of collection.

Reply: Ganoderma lucidum fruiting bodies were produced by Korean Ginseng Corporation company in Korea, they have developed smart cultivation system for commercial production of GL, we obtained commercially dried fruiting bodies in the month of Nov-Dec. So technically we are not involved in complete cultivation process of GL our main objective was to investigate dried GL.

  1. Figure 1 is not clear to the reader.

Reply: I tried to make it more clear but due to concentration of more detected compounds on one side its little difficult to address this issue but for clarification purpose I added supplementary table contain all detected compounds.

Reviewer 2 Report

The quality of this work is very high and it well presented 

there are few minor comments:

1- figure 3 need to be corrected, there is some overlap in the numbers. also can you please explain for the reader what do you mean by the red line cutting cross the bar charts?

2- figure 2, B it would be good for the reader if you explain different colors as green circle and read /blue hexagon

Author Response

  1. figure 3 need to be corrected, there is some overlap in the numbers. also can you please explain for the reader what do you mean by the red line cutting cross the bar charts?

Reply: In general, over than 70% of cell viability in MTT analysis can be accepted for non-cytotoxic effect. Therefore, we marked this point with a red line.

  1. figure 2, B it would be good for the reader if you explain different colors as green circle and read /blue hexagon

Reply: Done as suggested.

Reviewer 3 Report

General comments

English should be checked

The style used in the introduction should be improved (e.g Lines 56-65).

In table 01. SD values are very high and comparison could be biased.

Figures resolution should be improved.

The results were poorly reported.

The discussion should be improved by discussing the obtained results without including non useful bibliographic information. Moreover, the results should be compared with those previously reported in other studies.

A conclusion should be added

Specific comments

Section. Abstract

Remove abbreviations from the abstract

The abstract should be rewritten according to the IMRAD manner (a method subsection is missing)

Numerical values should be added to the results.

The PCA analysis results should be reviewed by a specialized reviewer. I am not able to confirm the statements based on this analysis.

Section. Introduction

Lines 48-50. Correct and complete the sentence.

Lines 56-58. Correct the sentence. Break down this long sentence into two correct sentences.

Lines 66-71. This paragraph should be removed since it is not adding a very useful information to the reader.

Line 75. Replace because of by owing to

Lines 83-115. This part should be rewritten in a more concise and short paragraph (it is too long and is about known facts related to inflammation and the roles of MAPK and NF-kappa B signalling pathways).

Lines 129-136. Replace by a simple direct and concise sentence about the main aim of the study, without details regarding methods.

Section. Results

Lines 139-152. Your results showed that heating resulted in higher content of organic acids. This should be clearly stated in the text.

Lines 148-152. It is stated that “The heat-dried method produced a higher level of amino acids than the freeze-dried method (L-Valine, L-Alanine, L-Serine, L-Asparagine, L-Threonine, and L-Proline) and monosaccharides (D-(-)- fructose, D-(+)- mannose, and D-(-)- ribofuranose). In the freeze- dried samples, organic acids were found to be significantly higher.”

By reading values from table 1, we can find that L-Valine content was 292,186 ± 64,67 at -80°C whereas heating gave a content of 2,214,288 ± 462,985. In the latter value, an SD of 462 is very high and I think it can be considered as a bias when making comparisons.

The same higher SD is found for L-Proline (66,414 ± 33,100)

Figure 2. the resolution should be improved.

Lines 188-205. Numerical values should be added to the results. The authors can compare at least two metabolites showing higher content with heating and two others possessing higher content with freezing.

Lines 220-222. This sentence should be corrected. We do not treat an extract with cells!!! We treat cells with GL extract.

Figure 3. The red line is inserted within the graph, why?

It seems that A and B do not have the same scale.

Line 247. Correct shows to showed

Figure 4A. I think there is an error between 2 and 10 (µg/ml) HDE. The results related to the concentration of  2 µg/ml should appear before those of 10 µg/ml.

The histograms should be improved and checked. Do they show the % of inhibition?

Section. Discussion

Lines 282-285. This conclusion should be removed since it is like a non-based generalization.

Lines 287-300. This should be removed since it has been included in the introduction.

The results are poorly discussed and are not compared with other previous studies.

Author Response

Section. Abstract

  1. Remove abbreviations from the abstract

Reply: Abbreviations removed from abstract as suggested. 

  1. The abstract should be rewritten according to the IMRAD manner (a method subsection is missing)

Reply: The abstract is thoroughly revised as suggested method subsection is improved.

  1. Numerical values should be added to the results.

Reply: modification done as suggested considering the fact we have limited number of words for abstract over all results section is improved as per recommendations.

  1. The PCA analysis results should be reviewed by a specialized reviewer. I am not able to confirm the statements based on this analysis.

Reply: Done as recommended.

Section. Introduction

  1. Lines 48-50. Correct and complete the sentence.

Reply: Done as suggested and marked in track changes.

  1. Lines 56-58. Correct the sentence. Break down this long sentence into two correct sentences.

Reply: Done as suggested and relevant references are added.

  1. Lines 66-71. This paragraph should be removed since it is not adding a very useful information to the reader.

Reply:  Irrelevant part is removed following the recommendations.

  1. Line 75. Replace because of by owing to

Reply: Done as suggested.

  1. Lines 83-115. This part should be rewritten in a more concise and short paragraph (it is too long and is about known facts related to inflammation and the roles of MAPK and NF-kappa B signalling pathways).

Reply: changes has been made according to recommendations.

  1. Lines 129-136. Replace by a simple direct and concise sentence about the main aim of the study, without details regarding methods.

Reply: statements improved with more focus on objective followed by methodology applied.

Section. Results

  1. Lines 139-152. Your results showed that heating resulted in higher content of organic acids. This should be clearly stated in the text.

Reply: Our GC-MS based analysis of freeze-dried samples shows significantly higher content of organic acids followed by heat dried samples having higher content of amino acids and monosaccharides.

  1. Lines 148-152. It is stated that “The heat-dried method produced a higher level of amino acids than the freeze-dried method (L-Valine, L-Alanine, L-Serine, L-Asparagine, L-Threonine, and L-Proline) and monosaccharides (D-(-)- fructose, D-(+)- mannose, and D-(-)- ribofuranose). In the freeze- dried samples, organic acids were found to be significantly higher.”

By reading values from table 1, we can find that L-Valine content was 292,186 ± 64,67 at -80°C whereas heating gave a content of 2,214,288 ± 462,985. In the latter value, an SD of 462 is very high and I think it can be considered as a bias when making comparisons.

The same higher SD is found for L-Proline (66,414 ± 33,100)

Reply: modifications has been done per recommendation. SD of L-Proline was 66,414 ± 33,100 in freeze dried samples as compared to heat dried samples.

  1. Figure 2. the resolution should be improved.

Reply: Added image file with higher resolution

  1. Lines 188-205. Numerical values should be added to the results. The authors can compare at least two metabolites showing higher content with heating and two others possessing higher content with freezing.

Reply: Done as suggested

  1. Lines 220-222. This sentence should be corrected. We do not treat an extract with cells!!! We treat cells with GL extract.

Reply: Done as recommended

  1. Figure 3. The red line is inserted within the graph, why?

Reply: In general, over than 70% of cell viability in MTT analysis can be accepted for non-cytotoxic effect. Therefore, we marked this point with a red line.

  1. It seems that A and B do not have the same scale.

Reply: Correction done as per suggestion.

  1. Line 247. Correct shows to showed

Reply: Done as suggested.

  1. Figure 4A. I think there is an error between 2 and 10 (µg/ml) HDE. The results related to the concentration of 2 µg/ml should appear before those of 10 µg/ml.

Reply: Reply: we sincerely apologize for our big mistake in presenting the wrong MTT assay results in figure 3. We found that HDE had weak toxicity at concentrations over 10 µg/ml compared to FDE through MTT assay. We carefully expect that this slight toxicity may hinder the manifestation of the effect of HDE on inflammation when it was induced by LPS. Thank you for your deep consideration again. And we changed the correct data in figure 3.

  1. The histograms should be improved and checked. Do they show the % of inhibition?

Reply: Thank you for your comment. As your suggestion, we added the inhibition rate in histogram figure 4.

Section. Discussion

  1. Lines 282-285. This conclusion should be removed since it is like a non-based generalization.

Reply: AS per suggestion mentioned portion was move to conclusion section with required modifications. Kindly refer to track changes for details.

  1. Lines 287-300. This should be removed since it has been included in the introduction.

Reply: Done as suggested marked by track changes.

Round 2

Reviewer 3 Report

Many thanks for the efforts made to improve the manuscript